# Sources of exposure to non-tobacco nicotine electronic nicotine delivery systems and associations with susceptibility to use and use behaviors among young adults in the United States

Wei Li[1]*, Grace Kong[1], Danielle R. Davis[1], Krysten W. Bold[1], Suchitra Krishnan-Sarin[1], Deepa R. Camenga[2], Meghan E. Morean[1]

1 Department of Psychiatry, Yale School of Medicine, New Haven, CT, United States of America,
2 Department of Emergency Medicine, Yale School of Medicine, New Haven, CT, United States of America

* Wei.vanness.li@yale.edu

## Abstract

### Introduction

The Electronic Nicotine Delivery Systems (ENDS) industry recently introduced non-tobacco nicotine (NTN), which is not tobacco-derived and is often marketed as "tobacco-free nicotine." Given its novelty, it is important to understand where young adults learn about NTN ENDS. This study examined sources of exposure to NTN ENDS and relationships with NTN ENDS use and susceptibility.

### Methods

We analyzed online survey data collected in Fall 2021 from 642 young adults (18–25 years) who had heard of NTN ENDS. We assessed 9 sources of NTN ENDS exposure (e.g., retail stores, social media) and examined associations between sources of exposure and NTN current (past-month) use, lifetime (non-current) use, and susceptibility to use, adjusting for demographics and other tobacco product use.

### Results

Participants reported current NTN ENDS use (37.4%), lifetime use (12.0%), susceptibility (18.5%), or no susceptibility to use (32.1%). The most common sources of NTN ENDS exposure were retail stores (87.7%) and social media (81.0%). Exposure to NTN ENDS via social media was associated with greater odds of current NTN ENDS use (vs. no susceptibility) (aOR = 1.83, 95%CI: 1.02–3.28). Exposure via online streaming platforms was associated with greater odds of current (aOR = 1.75, 95%CI: 1.08–2.82) and lifetime NTN ENDS use (aOR = 2.42, 95%CI: 1.25–4.68).

**Data Availability Statement:** All relevant data are within the manuscript and its Supporting Information files.

**Funding:** This work was supported by the American Heart Association (AHA) (20YVNR35460041), National Institute on Drug Abuse (NIDA) (K12DA000167) and US Food and Drug Administration (FDA) Center for Tobacco Products (CTP) (U54DA036151). Support for authors is also provided by R01DA049878 (GK), K01DA065494 (DD), and R01DA054993 (KB). The content is solely the responsibility of the authors and does not necessarily represent the official views of the AHA, the NIDA, or the FDA. The funders had no role in study design, data collection and analysis, decision to publish, or preparation of the manuscript.

**Competing interests:** The authors have declared that no competing interests exist.

## Conclusions

Young adults were exposed to and learned about NTN ENDS from diverse sources, primarily retail shops and social media. Further, exposure via social media and streaming platforms were associated with NTN ENDS use. Future studies should explore the content of NTN information from various sources to inform prevention efforts.

## Introduction

Electronic Nicotine Delivery Systems (ENDS) are the most popular tobacco product among young adults in the United States (US), with past-month use increasing from 2.4% in 2012 to 9.4% in 2020 [1, 2]. The rapid growth of ENDS use has sparked a global debate about their potential benefits or harm [3]. Reviews suggest that ENDS can assist smoking cessation among adults [4], while the long-term health effects of ENDS remain unclear and the nicotine in ENDS can harm the developing brain and lead to nicotine dependence among young people [5]. Therefore, tobacco regulations should consistently aim to prevent young people from initiating nicotine use.

The Family Smoking Prevention and Tobacco Control Act gave the US Food and Drug Administration (FDA) authority to regulate tobacco-derived nicotine (TDN) ENDS [6]. Subsequently, the ENDS industry introduced non-tobacco nicotine (NTN) or synthetic nicotine that is not tobacco-derived from tobacco [7]. The use of NTN ENDS is gaining popularity since 2021 [8]. Although national reports on the prevalence of NTN ENDS use are lacking, our earlier work, which involved preliminary self-report evidence from 1,176 young adults in the US, revealed that 49.2% had heard of NTN ENDS, 27.0% had ever tried them, and 29.6% were susceptible to trying them [9]. However, the introduction of NTN products poses a public health problem. NTN is often marketed as "tobacco-free nicotine" (TFN), a term that may mislead consumers into believing that NTN does not contain nicotine, reducing the perceived risks associated with NTN compared to TDN use, potentially increasing NTN product use among young adults [10]. However, limited research exists on NTN compares to TDN in terms of addictive potential, toxicity, and other harms. Although the FDA gained regulatory authority over NTN products in 2022 [7], these products continue to be freely marketed.

General ENDS-related content (e.g., opinions, reviews, news, advertisements) is readily accessible through a multitude of platforms and settings, including social media platforms [11], retail environments where ENDS products are sold, online forums and communities, as well as through targeted advertisements in both digital and traditional media [12, 13]. Exposure to ENDS-related content has been found to impact young people's use of ENDS and their susceptibility (susceptibility refers to the absence of a firm commitment to abstain from using substances, indicating a level of vulnerability to initiating the use of substances [14, 15]) towards these products [16–18]. In the context of general ENDS, being susceptible to use is associated with a greater likelihood of initiating and sustaining ENDS use [19]. Furthermore, emerging evidence suggests that youth NTN ENDS use is influenced by marketing that portrays NTN as a cleaner and lower-risk alternative to TDN [20]. However, little is known about where young adults learn about NTN ENDS, as well as which sources of exposure to NTN ENDS influence their use behaviors. Given the novelty of NTN tobacco products on the market, understanding this can inform tobacco regulatory efforts to reduce the appeal of NTN products to young adults. Therefore, we conducted an online survey of US young adults to examine self-reported sources of NTN ENDS exposure (e.g., retail stores, social media). We

hypothesized that exposure to each source would be associated with NTN ENDS current use, lifetime use, and susceptibility to use. Findings can inform surveillance of marketing practices and provide guidance of interventions or policies aimed at reducing NTN ENDS use among young people.

## Materials and methods

### Study design and procedures

The Yale University Institutional Review Board approved all study procedures (#1207010580). From September to October 2021, Qualtrics Online Sample (QOS), the secure market research division of *Qualtrics*, *Inc.*, recruited US young adults (18–25 years) to participate in a 20-minute, anonymous survey. Study eligibility included reporting on sex, race/ethnicity, and lifetime and past-month use of nicotine/tobacco products. Quotas ensured diversity in sex (approximately 50% female) and race/ethnicity (≤40% non-Hispanic White). Based on the aims of the larger project, quotas were set to ensure diversity in tobacco use status (i.e., never tobacco product use, lifetime tobacco product use without current use, current exclusive ENDS use, current exclusive use of non-ENDS tobacco products, and current use of both ENDS and other tobacco products), with ENDS use oversampled. Eligible participants provided written consent by marking a box prior to completing the full survey. Details of the study procedures can be found elsewhere [21].

### Measures

2,464 young adults initiated the survey. Participants were provided with a definition of "tobacco-free nicotine"—nicotine that is synthetic or artificial, meaning that it is created in a lab from chemicals that do not come from tobacco plants. Although the FDA recommends using the term "non-tobacco nicotine" (NTN) [22], we used "tobacco-free nicotine" in the survey due to its widespread commercial use. To ensure accurate understanding of the term tobacco-free nicotine, participants had to answer a comprehension question: "Based on the description you just read, which of the following is true about tobacco-free nicotine in vapes?" 1,424 participants (57.8%) correctly selected "Tobacco-free nicotine in vapes is synthetic or artificial and is created in a lab from chemicals that do not come from tobacco plants." Of these participants, 1,239 (87.0%) completed the entire survey.

### NTN ENDS awareness, use, and susceptibility

Participants were asked, "Before today, had you ever heard of tobacco-free nicotine vapes/ENDS?" (Yes/No). Participants who reported "No" were excluded. Those who responded "Yes" were asked, "Have you ever vaped tobacco-free nicotine, even one or two puffs?" (Yes/No). Those who responded "Yes" were categorized as ever using NTN ENDS. Those who ever used NTN ENDS were asked, "Approximately how many days out of the past 30 days did you vape tobacco-free nicotine?" Those who vaped NTN on ≥1 day were classified as currently using NTN ENDS. Those who reported trying NTN ENDS but indicated no past-month use were classified as lifetime use only.

Participants who had never vaped NTN were asked: 1) "How curious are you to try tobacco-free nicotine vapes?" ("not at all" to "very much") and 2) "At any time during the next year do you think you will use tobacco-free nicotine vapes?" ("definitely not" to "definitely yes"). These questions were adapted to NTN ENDS from prior research [15, 23]. Participants who responded anything other than "not at all" or "definitely not" to either question were considered susceptible [23].

## Sources of exposure to NTN ENDS

Participants reported how often they noticed NTN ENDS in the following platforms: retail stores (e.g., gas stations), billboards, social media (e.g., YouTube), non-social media websites (e.g., tobacco product websites), print media (e.g., newspapers), radio, network television, online streaming platforms (e.g., Netflix), and gaming platforms. Response options were dichotomized for each source into no ("never") versus yes ("rarely," "sometimes," and "often"). Finally, we summed all "yes" responses for exposure via each platform to reflect the total number of sources of exposure to NTN ENDS (range: 0–9 sources).

## Other tobacco product use

Participants reported if they had ever used heated tobacco, cigarettes, hookah, cigars/cigarillos, smokeless tobacco, or nicotine pouches. Responses for each item were "no" or "yes." Any endorsement of use was coded as using other tobacco products.

## Demographics

Participants' sex, age, and race/ethnicity were assessed. Race/ethnicity was coded as Non-Hispanic White, Non-Hispanic Black, Hispanic, and Non-Hispanic Other (including Asian, American Indian, Alaska Native, Native Hawaiian, Pacific Islander, and "Other").

## Statistical analyses

Statistical analyses were conducted using SAS 9.4 (SAS Institute, Cary, NC, USA). First, we performed descriptive statistics for all study variables. Second, chi-square and ANOVA tests (with Bonferroni corrections [$\alpha = 0.008$] for sources of exposure to NTN ENDS) were conducted to examine unadjusted differences in NTN ENDS use (i.e., current use, lifetime use, susceptible to use, not susceptible to use) based on sources of NTN exposure. Third, we used multinomial logistic regression to examine relationships between sources of exposure that were significant in the bivariate analyses and NTN ENDS use within the four groups. The non-susceptible group was set as the referent and covariates included sex, age, race/ethnicity, and non-vaping tobacco product use [20, 24]. The statistical significance level was set at $\alpha = 0.05$ for two-tailed tests. Multicollinearity was tested for the sources of NTN exposure prior to running the multinomial model by examining tolerance and Variance Inflation Factors (all values<2.5) using linear regression [25].

## Results

Our analytic sample (N = 642) was 55.1% female, 39.1% Non-Hispanic White, and had a mean age of 20.92 (SD = 2.28) years. 37.4% reported current NTN ENDS use, 12.0% reported lifetime use, 18.5% were susceptible to use, and 32.1% were not susceptible. 69.0% had used other tobacco products. Across all NTN ENDS use subgroups, the top two sources of NTN exposure were retail stores (87.7%) and social media (81.0%; **Table 1**).

Bivariate analyses showed that 6 out of 9 sources were significantly associated with four groups of NTN ENDS use (**Table 1**). Specifically, participants who reported current NTN ENDS use had the highest rates of NTN ENDS exposure via social media (87.5%) compared to those who indicated lifetime use (76.6%), susceptibility (84.0%), or no susceptibility to future use (73.3%). Furthermore, individuals who were susceptible to future use of NTN ENDS had the highest rates of exposure via print media (63.0%) and radio (58.8%) compared to those who endorsed current use (print media: 53.8%; radio: 43.8%), lifetime use (print media: 44.2%; radio: 31.2%), or no susceptibility to future use (print media: 40.8%;

**Table 1. Characteristics of the study population and differences in NTN ENDS use based on demographics and sources of exposure to NTN.**

| Variables | Analytic Sample n (%) (n = 642) | Tobacco-free nicotine (NTN) use status, n (%) | | | |
| --- | --- | --- | --- | --- | --- |
| | | Current use (n = 240) | Lifetime use (n = 77) | Susceptible to use (n = 119) | Non-susceptible to use (n = 206) |
| # Age (in years) | 20.92 (2.28) | 20.66 (2.14) [a] | 21.35 (2.33) [a] | 20.87 (2.34) [a] | 21.10 (2.35) [a] |
| Sex (female) ** | 354 (55.1) | 124 (51.7) [a] | 47 (61.0) [a] | 56 (47.1) [b] | 127 (61.7) [a] |
| Race/Ethnicity | | | | | |
| NH-White | 251 (39.1) | 101 (42.1) [a] | 37 (48.1) [a] | 34 (28.6) [a] | 79 (38.4) [a] |
| NH-Black | 120 (18.7) | 44 (36.7) [a] | 11 (14.3) [a] | 30 (25.2) [a] | 35 (17.0) [a] |
| Hispanic | 219 (34.1) | 78 (32.5) [a] | 26 (33.8) [a] | 41 (34.5) [a] | 74 (35.9) [a] |
| NH-Others | 52 (8.1) | 17 (7.1) [a] | 3 (3.9) [a] | 14 (11.8) [a] | 18 (8.7) [a] |
| Sources of exposure to NTN ENDS (Yes) | | | | | |
| Retail stores | 563 (87.7) | 217 (90.4) [a] | 68 (88.3) [a] | 104 (87.4) [a] | 174 (84.5) [a] |
| Billboards | 364 (56.7) | 140 (58.3) [a] | 40 (52.0) [a] | 72 (60.5) [a] | 112 (54.4) [a] |
| § Social media *** | 520 (81.0) | 210 (87.5) [a] | 59 (76.6) [b] | 100 (84.0) [b] | 151 (73.3) [b] |
| Non-social media websites *** | 412 (64.2) | 171 (71.3) [a] | 39 (50.7) [b] | 83 (69.8) [a] | 119 (57.8) [a] |
| Newspapers/magazines *** | 322 (50.2) | 129 (53.8) [a] | 34 (44.2) [a] | 75 (63.0) [b] | 84 (40.8) [a] |
| Radio *** | 274 (42.7) | 105 (43.8) [a] | 24 (31.2) [a] | 70 (58.8) [b] | 75 (36.4) [a] |
| Network television | 386 (60.1) | 151 (62.9) [a] | 44 (57.1) [a] | 79 (66.4) [a] | 112 (54.4) [a] |
| § Streaming platforms *** | 410 (63.9) | 172 (71.7) [a] | 51 (66.2) [a] | 79 (66.4) [a] | 108 (52.4) [b] |
| Gaming platforms *** | 307 (47.8) | 136 (56.7) [a] | 26 (33.8) [b] | 64 (53.8) [a] | 81 (39.3) [b] |
| # Total number of NTN ENDS sources ** | 5.78 (2.95) | 6.21 (2.70) [a] | 5.17 (2.76) [b] | 6.45 (2.89) [a] | 5.16 (3.18) [b] |
| Other tobacco product use (Yes) *** | 443 (69.0) | 181 (75.4) [a] | 58 (75.3) [a] | 84 (70.6) [a] | 120 (58.3) [b] |

Note

# Continuous data are presented as mean, standard deviation (M, SD) and categorical data are presented as sample size, percentage (n, %). Within rows, superscript letters reflect the results of the Bonferroni correction comparisons among the NTN ENDS use status (Chi-square for categorical variables; ANOVA for continuous variables). Within a row, cell values with different superscript letters assigned to them differ significantly from one another at $p < .008$ for sources of exposure to NTN ENDS and at $p < .05$ for other variables.

§ The original response options had four categories. The frequencies of each response option for social media were 19.0% (never), 23.7% (rarely), 36.3% (sometimes), and 21.0% (often), for streaming were 36.1% (never), 28.5% (rarely), 27.1% (sometimes), and 8.3% (often).

NH = Non-Hispanic, NTN = non-tobacco nicotine.

** $p < 0.05$

*** $p < 0.001$

radio: 36.4%). Individuals reporting current NTN ENDS use had higher rates of exposure via gaming platforms (56.7%) compared to those reporting lifetime use (33.8%) or no susceptibility (39.3%). Additionally, those reporting lifetime NTN ENDS use had the lowest rates of exposure via non-social media websites (50.7%) relative to the other three groups (current: 71.3%; susceptibility: 69.8%; non-susceptibility: 57.8%). Finally, individuals who reported no susceptibility to NTN ENDS use had the lowest rates of exposure via streaming platforms (52.4%) compared to the three other groups (current: 71.7%; lifetime: 66.2%; susceptibility: 66.4%).

In the adjusted model, exposure to NTN ENDS via social media was associated with greater odds of current NTN ENDS use (aOR = 1.83, 95%CI: 1.02–3.28; $p = 0.042$), and exposure to NTN ENDS via online streaming platforms was associated with greater odds of current and lifetime NTN ENDS use (current: aOR = 1.75, 95% CI: 1.08–2.82; $p = 0.023$; lifetime: aOR = 2.42, 95%CI: 1.25–4.68; $p = 0.009$; **Table 2**).

**Table 2. Multinomial logistic regression examining sources of exposure to NTN ENDS among young adults who reported current use, lifetime use and susceptibility to use NTN ENDS, in comparison to those who are not susceptible to use.**

| Variables | Adjusted Odds Ratio (95%CI) | | |
|---|---|---|---|
| | Current use [#] | Lifetime use [#] | Susceptible to use [#] |
| **Age (in years)** | **0.89 (0.82–0.98)** | 1.01 (0.90–1.14) | 0.95 (0.85–1.05) |
| **Male (Ref. = female)** [*] | **1.61 (1.08–2.40)** | 1.09 (0.62–1.89) | **1.93 (1.20–3.09)** |
| **Race/Ethnicity (Ref. = NH-White)** | | | |
| NH-Black | 0.94 (0.53–1.65) | 0.69 (0.31–1.54) | 1.67 (0.87–3.23) |
| Hispanic | 0.87 (0.55–1.38) | 0.86 (0.47–1.59) | 1.36 (0.76–2.41) |
| NH-Others | 0.79 (0.37–1.70) | 0.41 (0.11–1.52) | 1.86 (0.80–4.32) |
| **Sources of exposure to NTN (Ref. = No)** | | | |
| Social media | **1.83 (1.02–3.28)** | 1.19 (0.57–2.48) | 1.50 (0.76–2.98) |
| Non-social media websites | 1.18 (0.71–1.94) | 0.60 (0.31–1.18) | 0.89 (0.48–1.65) |
| Newspapers/magazines | 1.19 (0.72–1.97) | 1.44 (0.72–2.87) | 1.84 (1.00–3.37) |
| Radio | 0.71 (0.42–1.19) | 0.67 (0.32–1.39) | 1.61 (0.87–2.99) |
| Streaming platforms | **1.75 (1.08–2.82)** | **2.42 (1.25–4.68)** | 1.12 (0.63–2.00) |
| Games/multi-user gaming platforms | 1.42 (0.87–2.31) | 0.70 (0.35–1.38) | 0.97 (0.54–1.73) |
| **Other tobacco product use (Ref. = No)** | **2.63 (1.71–4.04)** | **2.29 (1.24–4.21)** | **1.85 (1.11–3.07)** |

[#] Non-susceptible to use is the reference group.

[*] Ref. represents reference group.

Note: Bolded point estimates indicate statistical significance at $p < .05$. The adjusted model did not include the total sources of exposure to NTN ENDS due to its collinearity with the sources of exposure to NTN ENDS.

## Discussion

This study is the first to investigate sources of exposure to NTN ENDS and their associations with NTN ENDS use behaviors among US young adults. The primary source of NTN ENDS exposure was retail stores, consistent with previous research showing that youth encounter general ENDS marketing in retail stores most frequently [15]. Retail stores such as gas stations or convenience stores provide easy access to NTN ENDS products and employ promotional strategies that may appeal to young adults. However, exposure to NTN ENDS in retail stores did not differ significantly based on ENDS use/susceptibility status, indicating a high level of exposure to NTN ENDS in retailers among young adults regardless of their use status. Prevention efforts to minimize exposure to NTN ENDS products in retail stores would be beneficial.

We found that social media was the second most frequent source of NTN ENDS exposure, and it was associated with current NTN ENDS use. Social media platforms are popular sources of marketing and offer novel marketing strategies not feasible in traditional marketing sources like TV and magazines [11]. For instance, influencers with substantial following on social media and a reputation in specific niches can promote NTN ENDS, potentially attracting a wide audience. Moreover, these platforms employ sophisticated algorithms to target ads to specific demographics (e.g., age, gender, interest) [26]. Research indicates that exposure to tobacco-related social media content fosters positive perceptions of tobacco products/brands among young people, possibly influencing tobacco use initiation or continuation [27]. NTN ENDS companies may exploit these tools to target young adults. To counter the pro-tobacco influence on social media, educational programs/campaigns that denormalize NTN ENDS use are necessary.

We also observed that exposure to NTN ENDS on streaming platforms was associated with current and lifetime NTN ENDS use. Streaming platforms are very popular among young

adults, with 90% opting to watch content on streaming over traditional TV/cable [28]. This popularity presents a novel source of tobacco content exposure including NTN ENDS, with 64% of "binge-worthy" shows on Netflix depicted tobacco use in 2021 [29]. As cigarette smoking declines and ENDS use rises, streaming platforms may increasingly depict vaping as more attractive and socially acceptable than smoking, potentially increasing ENDS use among young people [30]. As NTN ENDS grow in popularity [7], exposure to the NTN ENDS ads on diverse sources, including streaming platforms, may contribute to a rise in use. When considering the unique positioning of streaming platforms, it is crucial for streaming services to issue warnings about such content and modify their algorithms to reduce exposure to younger viewers, which may help mitigate the potential influence of streaming content on NTN ENDS use.

Several study limitations must be considered. First, the cross-sectional study design and online panel recruitment cannot be used to infer causality and may limit generalizability. Second, we relied on self-report data and could not verify the accuracy of participants' product use status. Third, considering the novelty of NTN information, even though we provided participants with a definition of TFN and assessed their comprehension of this term, it is possible that participants did not fully understand it and may have confused NTN with other terms, such as "nicotine-free", which were not assessed in the study. Fourth, individuals who reported using NTN ENDS may have been more likely to notice or seek out NTN information, leading to higher reported exposure to these products. Additionally, we did not assess specific sources of exposure to NTN (e.g., industry-supported ads for NTN products, discussions of NTN vaping among characters in movies/shows, mentions of NTN in the news) or how participants determined whether ENDS in these sources contained NTN vs. TDN and/or what context or information they used to make these determinations. Moreover, our original survey was conducted in response to initial reports of the presence of NTN in the marketplace and designed to assess the increase in NTN use. Therefore, we did not include a specific question regarding TDN ENDS use but instead asked about ENDS use in general, meaning that those who reported NTN use may also use TDN. Consequently, we could not determine how many participants exclusively used NTN ENDS or whether they had experience with TDN ENDS. Future studies should focus on comparing NTN and TDN ENDS use to provide insights for targeted regulations. Last, social media involves various platforms such as YouTube, Twitter, and Facebook. Exposure to each platform may have different impacts due to the mode and structure of the content. Considering the importance of social media in obtaining information among young adults, future studies should examine whether these different social media platforms communicate information about NTN differently. In summary, to address these limitations, future qualitative and quantitative research should explore NTN-related content on various platforms, including the use of probes or other questions to characterize the source and the content among young adults.

In conclusion, our study found that the top two sources of NTN ENDS exposure among young adults were retail stores and social media. Additionally, exposure to NTN ENDS via social media and streaming platforms was associated with NTN ENDS use behaviors. As the popularity of NTN products grows, it is important to maintain ongoing surveillance of NTN marketing strategies. To support FDA's regulation of NTN, it will be crucial to assess the specific content of advertisements, other forms of marketing messages (e.g., sponsored social media posts or events), and more subversive means of exposure (e.g., product placements in movies/TV shows, brand-related merchandise) to develop effective counter messaging to prevent NTN product use among young adults.

## Supporting information

**S1 Data.**
(XLSX)

## Author Contributions

**Conceptualization:** Grace Kong, Suchitra Krishnan-Sarin, Meghan E. Morean.

**Data curation:** Wei Li, Grace Kong, Danielle R. Davis, Krysten W. Bold, Suchitra Krishnan-Sarin, Deepa R. Camenga, Meghan E. Morean.

**Formal analysis:** Wei Li, Grace Kong, Meghan E. Morean.

**Funding acquisition:** Suchitra Krishnan-Sarin.

**Methodology:** Wei Li.

**Supervision:** Grace Kong, Suchitra Krishnan-Sarin, Meghan E. Morean.

**Visualization:** Grace Kong, Suchitra Krishnan-Sarin, Meghan E. Morean.

**Writing – original draft:** Wei Li.

**Writing – review & editing:** Wei Li, Grace Kong, Danielle R. Davis, Krysten W. Bold, Suchitra Krishnan-Sarin, Deepa R. Camenga, Meghan E. Morean.

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
