## [Decision Letter · Decision Letter 0]

24 Oct 2023

PONE-D-23-25277Sources of Exposure to Non-Tobacco Nicotine Electronic Nicotine Delivery Systems and Associations with Susceptibility to Use and Use Behaviors among Young Adults in the United StatesPLOS ONE

Dear Dr. Li,

Thank you for submitting your manuscript to PLOS ONE. After careful consideration, we feel that it has merit but does not fully meet PLOS ONE’s publication criteria as it currently stands. Therefore, we invite you to submit a revised version of the manuscript that addresses the points raised during the review process.

 As you can see, the reviewers have made several comments that need to be addressed before this manuscript can be published. Please address the key issues raised by Reviewer #1 in particular. You should expand the introduction to provide a more comprehensive background on non-tobacco nicotine (NTN) in electronic nicotine delivery systems (ENDS), its health effects, and its prevalence, given the significant increase in ENDS use, especially among adolescents and young adults. You should also consider the potential for confusion between NTN ENDS and nicotine-free ENDS (ENNDS) and clarify how this may impact the study's interpretation. Please also address the limitations of recall measures and provide more information on the source and structure of the content. Distinguishing between different modes of exposure and reporting frequency by each source will provide more context for the results. 

We look forward to receiving your revised manuscript.

Kind regards,

Daniel Demant, PhD, MPH, GradCertHEd, BAppSocSc

Academic Editor

PLOS ONE

Reviewers' comments:

Reviewer's Responses to Questions

**Comments to the Author**

1. Is the manuscript technically sound, and do the data support the conclusions?

Reviewer #1: Partly

Reviewer #2: Yes

2. Has the statistical analysis been performed appropriately and rigorously? 

Reviewer #1: Yes

Reviewer #2: Yes

3. Have the authors made all data underlying the findings in their manuscript fully available?

Reviewer #1: No

Reviewer #2: No

4. Is the manuscript presented in an intelligible fashion and written in standard English?

Reviewer #1: Yes

Reviewer #2: Yes

5. Review Comments to the Author

Reviewer #1: This paper reports the results of a cross-sectional survey in young adults reporting their exposure to non-tobacco nicotine ENDS and associations with use. Given the significant increases in ENDS use reported in in recent years, especially amount adolescents and young adults, as well as the well documented health impacts, this is timely research and provides important data on a topical public health issue. Generally the paper was well written and considers an important research question. I have a couple of comments for the authors where additionally clarity might be required:

1. I found the introduction a little brief especially given that non-tobacco nicotine has not been extensively investigated in the literature in relation to ENDS use in adolescents etc. I am not sure if it is a feature specific to the US context but there has been no mention of it in the e-cig related public health debate in Australia that I am aware of. Therefore, I was left with some questions about the health effects of NTN vs TDN, frequency of use of e-cigs with NTN vs TDN etc. What proportion of e-cigs use NTN vs TDN and how is this communicated to potential consumers? It would be helpful for the reader to have some more information on this.

2. The availability of “nicotine free” e-cigs (or ENNDS) is also used as a marketing tool by e-cig producers, with similar claims regarding being less risky/cleaner etc. I’m assuming you did not ask about ENNDS use in your survey, but I do wonder how many of your participants are fully cognisant of the differences between NTN e-cigs and nicotine free e-cigs. I appreciate the use of the comprehension question as an attempt to ensure understanding of the subject matter in participants, but given the way participants were recruited (an online survey panel), I am not fully convinced that participants understood what they were being queried about. I think this is probably worthy of some consideration when interpreting your findings (or consider as a potential limitation).

3. Per point #2, how feasible is it that any encounter with NTN ENDS via radio, tv, streaming platforms etc is a true encounter vs an encounter with ENDS more generally? For example, when seeing an actor vape on Netflix, I think it is unlikely that the viewer would be able to determine whether it was an NTN or TDN vape. Again, I think this is a limitation of the study, as I am not convinced the respondents were reporting at the same level of specificity as the question asked. I acknowledge that this is what you might be getting at when you write “we did not assess…how participants determined whether ENDS contained NTN vs TDN” , but I think this needs to be clearer.

4. Perhaps I somehow missed it in my reading, but did you assess how many of the participants had used/were susceptible to TDN ENDS use? Surely this is highly relevant to your findings and a factor that should be accounted for in your analysis. Or did you only include ENDS users in your analysis? Apologies if I have missed this but I couldn’t find this information anywhere in the methods (although you did mention oversampling ENDS users, but this is not the same thing)

5. In Table 1:

- According to the superscripts, the proportion of females in “not susceptible to use” was sigf different to the proportion of females in lifetime use (and the other categories). Is this correct as the proportions look very similar (61.7% vs 61%)

- The proportion in the “not susceptible to use” group who used other tobacco products seems very high (58.3%) given the relatively low rates of smoking etc in this age group. Although this is more understandable if you only included ENDS users.

6. When reporting the results of the bivariate analyses, I think it is somewhat misleading to imply that exposure via social media was sigf higher for current NTN ENDS users “compared to the other groups”. While this is the case for non-social media websites and gaming platforms, exposure via social media was only sigf higher compared to the non-susceptible group only.

Minor comments:

In the statistical analyses section, you mention Table 1 and the number of sources significant – this information should be taken out and presented in the results only.

Reviewer #2: This study aimed to identify sources of exposure to NTN (non-tobacco nicotine) for ENDS. The authors conducted a cross-sectional survey from 2021 among young adults (18-25). I have provided some questions and feedback that I hope are helpful to the authors.

Introduction

The authors write: “Exposure to general ENDS-related content (e.g., through retail stores, social media) has been found to impact young people’s ENDS use and susceptibility.” You might add a line or 2 here to explain what ENDS-related content is and what susceptibility means (e.g., susceptibility to what?). This would help with the next line about marketing, and the other content in the intro.

If space permits, there could be another line or 2 about where/what places/modes youth are exposed to ENDS content/marketing/advertising. It makes sense given the novelty that there would not be much research on exposure to NTN ENDS content, but there is plenty of information about exposure to ENDS content (generally) and ENDS use. Adding this info might help build rationale by (1) explaining what has been studied in related areas (but not NTN) and (2) identifying a gap that this paper could fill (related to NTN).

Methods/results

Ad/marketing exposure studies typically use (1) recall measures (do you recall seeing X?) and/or (2) recognition (here’s an example of X, have you seen this? If so, when and where?). It appears this study used recall measures. One limitation of recall measures is that participants might mistakenly recall something they did not see or misclassify something they saw. A good way to deal with these limitations is to include probes and other questions to characterize the source of the content and the structure of the content.

The measures for this study highlight a few related questions. For instance, did participants in this study see advertisements or marketing? Ads vs. marketing = two related but different forms of content. Did the content of NTN ENDS exposure come from industry sources (e.g., tobacco companies)? Friends in one’s social network? Were ENDS shown as part of narrative in a TV show? There should be more information to characterize the content. Was that information measured? Can the authors provide it?

There are related questions about mode, which touches on structure of the content. For instance, seeing videos on YouTube is different than seeing static text on Twitter, which is different than Facebook memes (and so on). Collapsing social media sites into 1 source would be like collapsing television and radio into the same category.

Can the frequency of exposure by social media site be reported? Same with streaming platforms? This would help contextualize results by source.

General comments:

I think the manuscript was well-written and the analyses were appropriate. The manuscript would be improved by providing more information about the sources of exposure. The authors claim that these data could help FDA regulate ENDS advertising/marketing on social media and streaming platforms, but it is difficult to see how these data could help if there is no information about the structure and source of the content.

I appreciate the chance to review, and I hope these comments are helpful: My best to the authors!

6. PLOS authors have the option to publish the peer review history of their article (what does this mean?). If published, this will include your full peer review and any attached files.

Reviewer #1: No

Reviewer #2: No

---

## [Author Response · Author response to Decision Letter 0]

15 Dec 2023

Reviewer #1: This paper reports the results of a cross-sectional survey in young adults reporting their exposure to non-tobacco nicotine ENDS and associations with use. Given the significant increases in ENDS use reported in in recent years, especially amount adolescents and young adults, as well as the well documented health impacts, this is timely research and provides important data on a topical public health issue. Generally the paper was well written and considers an important research question. I have a couple of comments for the authors where additionally clarity might be required:

Response: Thank you for your detailed review and suggestions. Our responses are outlined below.

1. I found the introduction a little brief especially given that non-tobacco nicotine has not been extensively investigated in the literature in relation to ENDS use in adolescents etc. I am not sure if it is a feature specific to the US context but there has been no mention of it in the e-cig related public health debate in Australia that I am aware of. Therefore, I was left with some questions about the health effects of NTN vs TDN, frequency of use of e-cigs with NTN vs TDN etc. What proportion of e-cigs use NTN vs TDN and how is this communicated to potential consumers? It would be helpful for the reader to have some more information on this.

Response and change: Thank you for providing suggestions on how to improve this manuscript. We have now mentioned the global e-cig debate and revised the introduction accordingly. Additionally, we have also added details regarding NTN vs. TDN, including frequency and communication to consumers. The revised Introduction now states:

“The rapid growth of ENDS use has sparked a global debate about their potential benefits or harm [1]. Reviews suggest that ENDS can assist smoking cessation among adults [2], while the long-term health effects of ENDS remain unclear and the nicotine in ENDS can harm the developing brain and lead to nicotine dependence among young people [3]. Therefore, tobacco regulations should consistently aim to prevent young people from initiating nicotine use.”

“The use of NTN ENDS is gaining popularity since 2021 [4]. Although national reports on the prevalence of NTN ENDS use are lacking, our earlier work, which involved preliminary self-report evidence from 1,176 young adults in the US, revealed that 49.2% had heard of NTN ENDS, 27.0% had ever tried them, and 29.6% were susceptible to trying them [5]. However, the introduction of NTN products poses a public health problem. NTN is often marketed as “tobacco-free nicotine” (TFN), a term that may mislead consumers into believing that NTN does not contain nicotine, reducing the perceived risks associated with NTN compared to TDN use, potentially increasing NTN product use among young adults [6]. However, limited research exists on whether NTN is comparable, safer, or less harmful than TDN. Although the FDA gained regulatory authority over NTN products in 2022 [7], these products continue to be freely marketed.”

2. The availability of “nicotine free” e-cigs (or ENNDS) is also used as a marketing tool by e-cig producers, with similar claims regarding being less risky/cleaner etc. I’m assuming you did not ask about ENNDS use in your survey, but I do wonder how many of your participants are fully cognizant of the differences between NTN e-cigs and nicotine free e-cigs. I appreciate the use of the comprehension question as an attempt to ensure understanding of the subject matter in participants, but given the way participants were recruited (an online survey panel), I am not fully convinced that participants understood what they were being queried about. I think this is probably worthy of some consideration when interpreting your findings (or consider as a potential limitation).

Response and change: Thank you for providing another great suggestion. Please note that we provided participants with a definition to help them understand what “tobacco-free nicotine” is and included a comprehension check question to help ensure participants had a shared understanding of NTN. We only analyzed data from those who correctly answered the comprehension check question. However, it is possible that some participants may still not have fully understood what NTN is. In addition, we did not ask about NTN relative to nicotine-free products, so we do not have data on a direct comparison. We believe this is a limitation of the study and we have included this in the limitations section. 

“Third, considering the novelty of NTN information, even though we provided participants with a definition of TFN and assessed their comprehension of this term, it is possible that participants did not fully understand it and may have confused NTN with other terms, such as “nicotine-free”, which were not assessed in the study.”

3. Per point #2, how feasible is it that any encounter with NTN ENDS via radio, tv, streaming platforms etc is a true encounter vs an encounter with ENDS more generally? For example, when seeing an actor vape on Netflix, I think it is unlikely that the viewer would be able to determine whether it was an NTN or TDN vape. Again, I think this is a limitation of the study, as I am not convinced the respondents were reporting at the same level of specificity as the question asked. I acknowledge that this is what you might be getting at when you write “we did not assess…how participants determined whether ENDS contained NTN vs TDN”, but I think this needs to be clearer.

Response and change: We appreciate your emphasis on this concern. We have now provided further explanation to clarify it and have also outlined the direction for future studies. 

“Additionally, we did not assess specific sources of exposure to NTN (e.g., industry-supported ads for NTN products, discussions of NTN vaping among characters in movies/shows, mentions of NTN in the news) or how participants determined whether ENDS in these sources contained NTN vs. TDN and/or what context or information they used to make these determinations. Moreover, our original survey was conducted in response to initial reports of the presence of NTN in the marketplace and designed to assess the increase in NTN use. Therefore, we did not include a specific question regarding TDN ENDS use but instead asked about ENDS use in general, meaning that those who reported NTN use may also use TDN. Consequently, we could not determine how many participants exclusively used NTN ENDS or whether they had experience with TDN ENDS.”

“In summary, to address these limitations, future qualitative and quantitative research should explore NTN-related content on various platforms, including the use of probes or other questions to characterize the source and the content among young adults.”

4. Perhaps I somehow missed it in my reading, but did you assess how many of the participants had used/were susceptible to TDN ENDS use? Surely this is highly relevant to your findings and a factor that should be accounted for in your analysis. Or did you only include ENDS users in your analysis? Apologies if I have missed this but I couldn’t find this information anywhere in the methods (although you did mention oversampling ENDS users, but this is not the same thing)

Response and change: We thank the reviewer for this perspective. In the survey, participants were initially asked if they had ever used ENDS in general, followed by a specific inquiry about TFN ENDS use. Everyone who had used TFN were included in the analytic sample. Since we did not directly inquire about past TDN use, it is not possible to determine if individuals in our sample used TDN ENDS in addition to TFN ENDS. The original survey was designed to respond to the rapid increase in NTN use, with a primary focus on NTN ENDS use behaviors rather than making direct comparisons between NTN and TDN ENDS use. Therefore, we did not assess the use or knowledge of TDN ENDS. The adjustment made in our analysis only accounted for other (non-ENDS) tobacco use as a covariate. We appreciate your suggestions, and we believe this would be a great direction for guiding future studies in investigating the comparison between TDN and TFN tobacco products. We added this to the limitations and it now states:

“Moreover, our original survey was conducted in response to initial reports of the presence of NTN in the marketplace and designed to assess the increase in NTN use. Therefore, we did not include a specific question regarding TDN ENDS use but instead asked about ENDS use in general, meaning that those who reported NTN use may also use TDN. Consequently, we could not determine how many participants exclusively used NTN ENDS or whether they had experience with TDN ENDS. Future studies should focus on comparing NTN and TDN ENDS use to provide insights for targeted regulations.”

5. In Table 1:

5a. According to the superscripts, the proportion of females in “not susceptible to use” was sigf different to the proportion of females in lifetime use (and the other categories). Is this correct as the proportions look very similar (61.7% vs 61%)

Response and change: We appreciate you pointing out the error. The superscript letter should be in the “susceptible to use” group to indicate a difference. We have now corrected it. Additionally, we reviewed the entire Table 1 and found the significant level for the sources of exposure variable (only) should be set at 0.008 (Bonferroni correction) rather than 0.05. We have amended the table accordingly and included asterisk signs to make it clearer. 

5b. The proportion in the “not susceptible to use” group who used other tobacco products seems very high (58.3%) given the relatively low rates of smoking etc in this age group. Although this is more understandable if you only included ENDS users.

Response: We appreciate the reviewer’s thoughtful comments. In our study, the 58.3% represents those who have never used ENDS but have tried other tobacco products such as cigarettes. While this may seem high, it is understandable given that the participants in this age group often experiment with different tobacco products. It is important to note that the 58.3% refers to susceptibility among those with lifetime or ever past use of other tobacco products, not current or past-month use of these products.

6. When reporting the results of the bivariate analyses, I think it is somewhat misleading to imply that exposure via social media was sigf higher for current NTN ENDS users “compared to the other groups”. While this is the case for non-social media websites and gaming platforms, exposure via social media was only sigf higher compared to the non-susceptible group only.

Response and change: We appreciate the suggestion. We have expanded the results of the bivariate analyses to provide a clearer and more detailed report. 

“Bivariate analyses showed that 6 out of 9 sources were significantly associated with four groups of NTN ENDS use (Table 1). Specifically, participants who reported current NTN ENDS use had the highest rates of NTN ENDS exposure via social media (87.5%) compared to those who indicated lifetime use (76.6%), susceptibility (84.0%), or no susceptibility to future use (73.3%). Furthermore, individuals who were susceptible to future use of NTN ENDS had the highest rates of exposure via print media (63.0%) and radio (58.8%) compared to those who endorsed current use (print media: 53.8%; radio: 43.8%), lifetime use (print media: 44.2%; radio: 31.2%), or no susceptibility to future use (print media: 40.8%; radio: 36.4%). Individuals reporting current NTN ENDS use had higher rates of exposure via gaming platforms (56.7%) compared to those reporting lifetime use (33.8%) or no susceptibility (39.3%). Additionally, those reporting lifetime NTN ENDS use had the lowest rates of exposure via non-social media websites (50.7%) relative to the other three groups (current: 71.3%; susceptibility: 69.8%; non-susceptibility: 57.8%). Finally, individuals who reported no susceptibility to NTN ENDS use had the lowest rates of exposure via streaming platforms (52.4%) compared to the three other groups (current: 71.7%; lifetime: 66.2%; susceptibility: 66.4%).”

7. Minor comments:

In the statistical analyses section, you mention Table 1 and the number of sources significant – this information should be taken out and presented in the results only.

Response and change: This is a helpful comment. We have moved this information to the beginning of the bivariate results section. 

Reviewer #2: This study aimed to identify sources of exposure to NTN (non-tobacco nicotine) for ENDS. The authors conducted a cross-sectional survey from 2021 among young adults (18-25). I have provided some questions and feedback that I hope are helpful to the authors.

Response: Thank you for your kind words. We appreciate your overall positive assessment.

Introduction

1. The authors write: “Exposure to general ENDS-related content (e.g., through retail stores, social media) has been found to impact young people’s ENDS use and susceptibility.” You might add a line or 2 here to explain what ENDS-related content is and what susceptibility means (e.g., susceptibility to what?). This would help with the next line about marketing, and the other content in the intro.

Response and change: We are thankful for the opportunity to clarify. We have revised the introduction and it now states: 

“General ENDS-related content (e.g., opinions, reviews, news, advertisements) is readily accessible through a multitude of platforms and settings, including social media platforms [8], retail environments where ENDS products are sold, online forums and communities, as well as through targeted advertisements in both digital and traditional media [9, 10]. Exposure to ENDS-related content has been found to impact young people’s use of ENDS and their susceptibility (which refers to the absence of a firm commitment to abstain from using substances, indicating a level of vulnerability to initiating the use of substances [11,12]) to using these products [13-15].”

2. If space permits, there could be another line or 2 about where/what places/modes youth are exposed to ENDS content/marketing/advertising. It makes sense given the novelty that there would not be much research on exposure to NTN ENDS content, but there is plenty of information about exposure to ENDS content (generally) and ENDS use. Adding this info might help build rationale by (1) explaining what has been studied in related areas (but not NTN) and (2) identifying a gap that this paper could fill (related to NTN).

Response and change: Thank you for this helpful comment. To comprehensively address this, we have incorporated the relevant information in the following:

“General ENDS-related content (e.g., opinions, reviews, news, advertisements) is readily accessible through a multitude of platforms and settings, including social media platforms [8], retail environments where ENDS products are sold, online forums and communities, as well as through targeted advertisements in both digital and traditional media [9, 10].”

“In the context of general ENDS, being susceptible to use is associated with a greater likelihood of initiating and sustaining ENDS use [16].”

Methods/results

3. Ad/marketing exposure studies typically use (1) recall measures (do you recall seeing X?) and/or (2) recognition (here’s an example of X, have you seen this? If so, when and where?). It appears this study used recall measures. One limitation of recall measures is that participants might mistakenly recall something they did not see or misclassify something they saw. A good way to deal with these limitations is to include probes and other questions to characterize the source of the content and the structure of the content.

Response and change: This is a helpful suggestion. We have now further clarified this in the limitations and revised the content accordingly. We also included this in the guidance for the future studies. 

“Additionally, we did not assess specific sources of exposure to NTN (e.g., industry-supported ads for NTN products, discussions of NTN vaping among characters in movies/shows, mentions of NTN in the news) or how participants deter

---

## [Decision Letter · Decision Letter 1]

13 Feb 2024

Sources of Exposure to Non-Tobacco Nicotine Electronic Nicotine Delivery Systems and Associations with Susceptibility to Use and Use Behaviors among Young Adults in the United States

PONE-D-23-25277R1

Dear Dr. Li,

We’re pleased to inform you that your manuscript has been judged scientifically suitable for publication and will be formally accepted for publication once it meets all outstanding technical requirements.

Kind regards,

Daniel Demant, PhD, MPH, GradCertHEd, BAppSocSc

Academic Editor

PLOS ONE

Additional Editor Comments (optional):

Reviewers' comments:

Reviewer's Responses to Questions

**Comments to the Author**

1. If the authors have adequately addressed your comments raised in a previous round of review and you feel that this manuscript is now acceptable for publication, you may indicate that here to bypass the “Comments to the Author” section, enter your conflict of interest statement in the “Confidential to Editor” section, and submit your "Accept" recommendation.

Reviewer #1: All comments have been addressed

Reviewer #2: All comments have been addressed

2. Is the manuscript technically sound, and do the data support the conclusions?

Reviewer #1: Yes

Reviewer #2: Yes

3. Has the statistical analysis been performed appropriately and rigorously? 

Reviewer #1: Yes

Reviewer #2: Yes

4. Have the authors made all data underlying the findings in their manuscript fully available?

Reviewer #1: Yes

Reviewer #2: No

5. Is the manuscript presented in an intelligible fashion and written in standard English?

Reviewer #1: Yes

Reviewer #2: Yes

6. Review Comments to the Author

Reviewer #1: I thank the authors for considering my previous comments. They have all been adequately addressed. I congratulate the authors on this well written paper which I think is a valuable contribution to the literature.

Reviewer #2: I appreciate the changes and responses to reviewer comments.

7. PLOS authors have the option to publish the peer review history of their article (what does this mean?). If published, this will include your full peer review and any attached files.

Reviewer #1: No

Reviewer #2: No

---

## [Editor Report · Acceptance letter]

26 Apr 2024

PONE-D-23-25277R1 

PLOS ONE

Dear Dr. Li, 

I'm pleased to inform you that your manuscript has been deemed suitable for publication in PLOS ONE. Congratulations! Your manuscript is now being handed over to our production team.

Kind regards, 

on behalf of

Dr. Daniel Demant 

Academic Editor

PLOS ONE